# Synergistic Action of Gefitinib and GSK41364A Simultaneously Loaded in Ratiometrically-Engineered Polymeric Nanoparticles for Glioblastoma Multiforme

**DOI:** 10.3390/jcm8030367

**Published:** 2019-03-15

**Authors:** Praveena Velpurisiva, Prakash Rai

**Affiliations:** 1Department of Biomedical Engineering and Biotechnology, University of Massachusetts Lowell, Lowell, MA 01854, USA; praveena_velpurisiva@student.uml.edu; 2Department of Chemical Engineering, University of Massachusetts Lowell, Lowell, MA 01854, USA

**Keywords:** combination therapy, cancer, glioblastoma multiforme, polymeric nanoparticles, gefitinib, GSK461364A, drug resistance, synergistic effect, drug interaction, enhanced permeation and retention

## Abstract

Glioblastoma Multiforme is a deadly cancer of glial cells with very low survival rates. Current treatment options are invasive and have serious side effects. Single drug treatments make the tumor refractory after a certain period. Combination therapies have shown improvements in treatment responses against aggressive forms of cancer and are becoming a mainstay in the management of cancer. The purpose of this study is to design a combinatorial treatment regimen by engineering desired ratios of two different small molecule drugs (gefitinib and GSK461364A) in a single carrier that can reduce off-target effects and increase their bioavailability. Synergistic effects were observed with our formulation when optimal ratios of gefitinib and GSK461364A were loaded in poly (lactic-co-glycolic) acid and polyethylene glycol (PLGA-PEG) nanoparticles and tested for efficacy in U87-malignant glioma (U87-MG) cells. Combination nanoparticles proved to be more effective compared to single drug encapsulated nanoparticles, free drug combinations, and the mixture of two single loaded nanoparticles, with statistically significant values at certain ratios and drug concentrations. We also observed drastically reduced clonogenic potential of the cells that were treated with free drugs and nanoparticle combinations in a colony forming assay. From our findings, we conclude that the combination of GSK461364A and higher concentrations of gefitinib when encapsulated in nanoparticles yield synergistic killing of glioma cells. This study could form the basis for designing new combination treatments using nanoparticles to deliver multiple drugs to cancer cells for synergistic effects.

## 1. Introduction

Glioblastoma Multiforme (GBM) is the cancer of glial cells that rapidly spreads to various areas of the brain [1]. Of all the known malignant forms of cancers diagnosed in the central nervous system, GBM has a major share of 81% [2]. This aggressive form of cancer is more often seen in white male patients above 70 years of age than in women and other ethnicities, while some forms of astrocytomas are seen in children [3]. The number of anticancer drugs that are clinically approved for GBM remain low due to the tumor heterogeneity [4]. Novel therapies such as tumor-treating fields (TTF) have been approved to treat newly diagnosed GBM patients where the alternating electrical signals eventually block cell division and proliferation [5]. In the EF-14clinical trial, TTF when combined with temozolomide, and NovoTTF-100A, were proven to show significant improvement in overall survival of recurrent GBM patients and also resulted in higher progression free survival [6,7]. Although the technological advancements claim that there is a decrease in mortality rate among cancer patients, 2018 statistics point out that GBM still remains incurable with a median survival of 15 months [8]. Surgery proves to be a better solution in those cancers where tissues can be safely resected without affecting the organ function [9], but in cancers of brain, it may impair the cognitive functions [10]. Although the tumors are identified using stereotactical biopsies and positron emission tomography (PET) scans, indicating areas of higher metabolic activity, 35–40% of GBM patients have unresectable tumors that are deeply embedded in the brain, leaving such patients with chemotherapy and radiotherapy [11,12]. 

Deeper understanding of molecular pathways is crucial to identify the targets that are derailed in cancers and therefore helps us choose suitable therapeutic agents. Several targeted agents have emerged to minimize the toxic effects towards the healthy tissues. Forty percent of GBM cases report mutations in epidermal growth factor receptor (EGFR) [13], namely EGFR onco-variant III (EGFR vIII) [14]. EGFRs play a major role in neuronal, epithelial, and mesenchymal tissues in cellular development, specifically proliferation and differentiation [15]. In EGFR vIII mutation, the receptor–ligand binding is not required to trigger the pathway downstream as seen in a normal pathway, but the tyrosines flanking either side of receptors are constitutively autophosphorylated leading to more cell proliferation. Tyrosine kinase inhibitors (TKIs) have shown promising anti-cancer effects in treating different cancers [16]. Gefitinib is one such small molecule TKI that was approved for the treatment of non-small cell lung cancer (NSCLC) [17]. Gefitinib competes with the Adenosine Triphosphate (ATP) binding site and blocks the ATP driven tyrosine phosphorylation that is crucial for epidermal growth factor (EGFR) signaling and this action is independent of blocking the receptor [18]. Gefitinib dephosphorylates Bcl-2 associated death promoter (BAD) and triggers cell death via apoptosis [19].

Cell cycle machinery is deregulated in cancers and Polo-like kinases 1 (PLK-1s) are crucial at different stages of cell cyle progression such as spindle formation, transition to mitotic phase, and cytokinesis in a normal cell [20]. Since the cell proliferation is higher in malignant cells, PLK1s are profoundly expressed in rapidly dividing cells [21]. PLK-1 inhibitors have shown favorable anti-tumorigenic effects [22]. Of these inhibitors, we chose GSK461364A for our study due to its higher specificity to PLK-1. GSK461364A induces cell death by causing cell cycle arrest at G2/M phase by competing with the ATP binding site of PLK-1 thus leading to apoptosis [23].

Despite these, there is sufficient evidence in the literature that suggests the malignant cells becoming refractory to monotherapy in cancer patients [24]. Due to the quagmire of signaling pathways in cancers, suppression of a single cellular pathway by a single therapeutic agent is not sufficient as the malignant cells continue to proliferate by deregulating other pathways. Hence combinatorial regimen is effective by blocking multiple vital components of the cell machinery simultaneously. Combination treatment is preferred to sequential treatment in the case of aggressive cancers such as GBM, as the tumor rapidly spreads and needs to be curtailed [25]. Simultaneously interfering with multiple non-parallel targets in the dysregulated cancer signaling pathways can be more effective than sequential treatments [26]. Zheng et al. proved a significant reduction in tumor burden with their dual drug loaded nanoparticles targeting EGFR and MET pathways simultaneously [27]. Synergistic, additive, or antagonistic effects on the cells in vitro are subject to the ratios and concentrations in which the therapeutic agents are administered [28]. Certain drug combinations such as temozolomide and lomustine have increased the overall survival by six months in phase III clinical trials in newly diagnosed GBM patients with O6-Methylguanine-DNA Methyltransferase (MGMT) methylation [29].

The combination of these small molecule drugs will have a significant impact on at least 40% of GBM patients post evaluation. The mutated receptors such as EGFR vIII are upregulated in GBM and hence result in resistance to the therapies that target the ligands that bind to EGFR. Ratiometric control is crucial for a combinatorial therapy because administration of a mixture of free drugs may result in exposure of cells to undesirable ratios resulting in poor efficacy due to antagonistic effect. Simultaneously administering a mixture of free drugs can also compound their off-target toxicities due to drug–drug interactions [30]. Nanotechnology offers a novel way for combination treatments by simultaneously delivering multiple drugs to cancer cells with improved efficacy and reduced toxicities [31]. Nanotechnology enables designing drug delivery vehicles of required size and surface charge that are some of the crucial physicochemical properties to be considered for drug delivery across physical barriers in the body [32]. Nanomedicine enables engineering of drug carriers with sustained drug release and is important in encapsulating drugs of a set defined ratio, as described in our study. It provides a leeway in adjusting the composition of the polymers based on the objective of the study. Nanomedicines are not just specific to studies in cancer research but widely used in diagnostics and therapeutics pertaining to Parkinson’s [33], cardiovascular [34], chronic kidney disease [35], etc.

Polymeric nanoparticles (NPs) are approved organic nanocarriers that can entrap hydrophobic drugs in the required ratios and offer controlled drug release under different conditions [36]. In our study, we engineered NPs to encapsulate precise ratios of two drugs (Figure 1) and tested them at effective concentrations resulting in synergistic killing of GBM cells. NPs can increase the therapeutic index of the payloads administered and the basis for this argument stems from the higher cellular uptake as demonstrated in our previous study [37], and enhanced retention and cytotoxicity observed with the drug encapsulated nanoparticles as shown in our previous study [37]. The size of the nanoparticles is a crucial factor for drug delivery systems that are expected to cross the blood brain barrier (bbb). In our experiments, we ensured the size to be sub 200 nm that is ideal for the entry across bbb [38]. When tested in vivo, NPs can be designed to provide uniform biodistribution of both the drugs in the tissues. To enhance the cytotoxicity of the drugs and reduce the survival of GBM cancer cells, we demonstrate that the combination of gefitinib and GSK461364A, encapsulated in poly (lactic-co-glycolic) acid and polyethylene glycol (PLGA-PEG) nanoparticles, promotes additive or synergistic effects in U87-MG cells, containing highly aberrant chromosomes [39]. This co-encapsulation strategy offers ratiometric control over loading of both the drugs in the desired ratios. PEG enhances the stability of the formulation in vitro and avoids quicker clearance by the mononuclear phagocytic system (MPS) in vivo [40]. This robust ratiometric combinatorial approach will be enhanced when encapsulated in PLGA-PEG NPs that can prevent rapid drug degradation and clearance from the system and thereby increase the bioavailability of the drugs.

Figure 1 illustrates the mechanism of action of both the drugs when entrapped in polymeric nanoparticles and their effect when tested in glioma cells (U87-MG). Both the small molecule drugs in their free form, or when encapsulated in NPs, will bind to the plasma membrane and are internalized through endocytosis. With two different mechanisms of action, namely tyrosine kinase inhibitory activity of gefitinib and mitotic arrest at G2/M phase by GSK461364A, an accelerated cell death of glioma cells is observed with a synergistic effect of the combination. To meet the needs of new and effective combination therapies, our study adds a vital dimension to the treatment possibilities of GBM.

## 2. Experimental Section

Poly(lactide-co-glycolide) (PLGA) (MW: 10,000–15,000 Da), Methoxy poly(ethylene glycol)-b-poly(lactide-co-glycolide) (mPEG-PLGA) (MW: 2–15 k.Da) were purchased from PolySciTech^®^ (West Lafayette, IN, USA). Gefitinib was purchased from Cayman Chemicals (Ann Arbor, MI, USA) and reconstituted in methanol; GSK461364A was purchased from APExBIO (Houston, TX, USA) and was dissolved in ethanol; (3-(4,5-dimethylthiazol-2-yl)-5-(3-carboxymethoxyphenyl)-2-(4-sulfophenyl)-2H-tetrazolium) MTSassay kit was purchased from Promega Corporation^®^ (Madison, WI, USA), Triton-X and other HPLC grade organic solvents were obtained from Fisher Scientific™ (Agawam, MA, USA). Human GBM cell line, U87-MG was bought from ATCC^®^ (Manassas, VA, USA). Eagle’s minimum essential (MEM) alpha modification media with L-glutamine (Genclone™) with 10% fetal bovine serum (FBS) (Genclone™, San Diego, CA, USA) and 1% PenStrep (Gibco™, Fisher Scientific) was used to culture and maintain the cells at 37 °C, 5% CO_2_. Crystal Violet was purchased from Sigma-Aldrich (St. Louis, MO, USA). 

### 2.1. Ratiometric Synthesis and Characterization of Co-Loaded PLGA-PEG Nanoparticles

Amphiphilic co-polymers of PLGA and PEG were used to prepare PLGA-PEG NPs with a blend of 75:25 of PLGA and PLGA-PEG based on our previous study [37]. Dual drug loaded NPs were synthesized by adding gefitinib:GSK461364A in the molar ratio of 4.54:1. This ratio was determined based on the results from cytotoxicity assays with the free drug combinations tested in U87-MG cells as shown in Appendix A and explained in Section 2.5. The absorbance of gefitinib was measured at 331 nm and GSK461364A at 272 and 305 nm using Nanodrop™ 2000c Ultraviolet-Visible (UV-Vis) spectrophotometer (Thermo Fisher Scientific, Delaware City, DE, USA). Both the drugs were ratiometrically added and these combination NPs were synthesized using the nanoprecipitation method [41]. Similarly, empty NPs (without the drugs), only gefitinib (gef NP) or only GSK461364A (GSK461364A NP) with the same concentration as the combination NPs containing gefitinib and GSK461364A (combo NP) were prepared. 

#### 2.1.1. Characterization of Co-Encapsulated Nanocarriers

Once the organic solvent evaporated, the nanoparticles formed as a result of self-assembly were purified via centrifugation at 800× *g* for 10 min at 25 °C using 30KDa Amicon centrifugal tubes purchased from Millipore Sigma (Burlington, MA, USA). These nanoparticles were characterized for their size, surface charge, and polydispersity index (PDI) using Malvern Zetasizer Nano ZS90 purchased from Malvern Analytical Inc. (Westborough, MA, USA). The graphic output of the size distribution and the zeta potential are shown in Appendix A.

#### 2.1.2. Morphological Characterization 

The morphology of combo NPs was examined by 120 kV transmission electron microscope (TEM) (Philips EM-400T). A sample volume of 2–3 µL of diluted nanoparticles were added on carbon 200 mesh, copper (Electron microscopy sciences), air dried for 48 h and observed using TEM. 

Surface information of NPs was obtained by performing field emission scanning electron microscopy (SEM) (JEOL JSM 7401F), where a similar volume of sample was added on a silicon wafer. The sample was allowed to air dry for 48 h and sputter coated with gold using vacuum sputter coater (Denton Vacuum Desk IV, Moorestown, NJ, USA) for good electron conductivity and observed under SEM.

#### 2.1.3. Determination of Encapsulation Efficiency

To determine the amount of drug encapsulated in single drug loaded NPs, empty nanoparticles containing 1% Triton-x was used as blank. The concentration of gef NPs was calculated by measuring the absorbance of gefitinib at 331 nm using Nanodrop™ 2000c UV-Vis spectrophotometer (Thermo Fisher Scientific, Delaware City, DE, USA). Likewise, the concentration of GSK461364A NPs was calculated by their absorbance at 311 nm. To determine the amount of both the drugs encapsulated in the combo NPs, we used the following method. The concentration of GSK461364A in combo NPs was determined by blanking with gef NPs and similarly the concentration of gefitinib in combo NPs was measured by blanking with GSK461364A NPs. The absorbance peaks of the combo NPs are shown in Appendix A. Although the wavelengths of both the drugs are not far apart, the absorbance values were confirmed using High Performance Liquid Chromatography (HPLC) (Agilent 1100) as explained in the Section 2.4.1. Percentage of encapsulation of drugs was calculated as follows.
% Drug encapsulation efficiency = (Amount of drug in mg upon characterization/Amount of drug added during synthesis) × 100(1)

### 2.2. In Vitro Stability of Co-Loaded PLGA-PEG Nanoparticles

In vitro stability of combo NPs was investigated by storing the formulation at 4 °C and in media containing 10% FBS at 37 °C to monitor their stability for shelf life and at a physiologically relevant temperature.

### 2.3. Stability of Free vs. Nano Drug in Media Containing Fetal Bovine Serum

In vitro stability of gefitinib in free form (free gef), GSK461364A in free form (free GSK461364A) was compared against the stability of gef NPs, GSK461364A NPs (of same concentrations as free drugs) and combo NPs by incubating the same concentration of each of the drugs in MEM alpha media containing 10% FBS. The absorbance readings were measured at different time intervals until 48 h and the decrease in concentration of drugs was plotted against time.

### 2.4. Measurement of In Vitro Drug Release from the Combination Nanoparticles

In vitro drug release of both the drugs from PLGA-PEG NPs was evaluated by performing the dialysis bag diffusion method. Well characterized sample was added to a regenerated cellulose dialysis bag with a MWCO of 20 k.Da (Spectra Max^®^, Chicago, IL, USA). This dialysis bag was placed in a beaker containing 800 mL of 1X PBS and maintained at 37 °C, 150 rpm. Samples from the dialysis bag were collected from the beaker periodically. To clearly understand the release profiles of both the drugs, the samples were analyzed using reverse phase high performance liquid chromatography (HPLC).

#### 2.4.1. Peak Separation Using Reverse Phase HPLC

Samples were added with 1% Triton-x to break open the nanoparticles and 50 µL of this mixture was injected into Agilent 1100 High Performance Liquid Chromatography (Waldbronn, Germany) with C18 reverse phase column as stationary phase and 0.1% TFA in acetonitrile as mobile phase. Samples were injected with a flow rate of 1 mL/min with a total run time of 16 min. Peak eluents of the signals detected at 331 nm, 226 nm, 295 nm, and 311 nm at different time points were collected.

#### 2.4.2. Peak Confirmation Using Mass Spectrometer

Shimadzu 8040 Triple Quadrupole Liquid chromatograph mass spectrometer (LC/MS) (Columbia, MD, USA) was used to confirm the molecular weights of the peak eluents and clearly distinguish the peaks arising due to either gefitinib or GSK461364A thereby determining the amount of drug present in the sample at different intervals. Once the samples were confirmed, the drug release profiles were plotted using DDSolver [42].

### 2.5. EC_50_ Determination of Gefitinib and GSK461364A

Effective concentration (EC_50_) of both the drugs were determined by treating 10,000 U87-MG cells per well in a 96-well plate with a wide range of concentrations of free gefitinib and half the wells in the plate with GSK461364A. Cell viability was assessed by performing standard MTS assay. Appropriate live and dead controls were used for this study. EC_50_ of GSK461364A was clearly explained in our previous study [37]. EC_50_ curve for gefitinib is 65.9 µM as shown in Appendix A. 

### 2.6. Estimation of the Drug Concentrations in the Ratiometric Study from the Free Drug Combination- Cytotoxicity Study

After determining EC_50_ values of each of the drugs, various combinations of different concentrations of gefitinib and GSK461364A were tested in U87-MG cells and the cell viabilities were determined as shown in Appendix A. The results from this experiment helped us to determine the ratios of the drugs that we used in the synthesis process.

### 2.7. Measurement of Cytotoxicity-Evaluation of the Treatment for Synergy in Monolayer Cultures of U87-MG

Cells were seeded with cell density of 10,000 cells per well in a 96-well cell culture plate. Once the cells were well adhered to the dish, they were dosed with various concentrations of free gef, nano gef, free GSK461364A, nano GSK461364A, a combination of free gef and free GSK461364A (free combo), combination of nano gefitinib and nano GSK461364A, and combo NPs containing gefitinib and GSK461364A in the same carrier. Controls for this experiment included cells that received no treatment and dead control (cells added with deionized water to induce osmotic shock). Once the cells were incubated with these formulations for 72 h at 37 °C, 5% CO_2_, they were added with 20 µL of activated MTS reagent to a final volume of 100 µL. After 90 min, the absorbance was read at 490 nm using spectrophotometer (Spectramax M2e, Molecular Devices).

### 2.8. Clonogenic Assay

To evaluate the clonogenic potential of the cells post treatment, U87-MG cells were plated at a density of 1000 cells per well in a six well plate. Once the cells were adhered, they were dosed with free gef, free GSK461364A, nano gef, nano GSK461364A, free combo, combination of nano gefitinib and nano GSK461364A, combo NPs. Control for this experiment included cells that received no treatment. The cells were allowed to grow for 10 days and the protocol by Yang et al. [43] was used to process the plates. Colonies were counted using an inverted digital light microscope (EVOS™ XL Core Imaging System) with colony defined as a group of cells of 50 or more. The colonies thus formed were quantified using the formula
% Cell survival rate = (Number of colonies in treated plates/Number of colonies in No treatment (control)) × 100(2)

## 3. Results

### 3.1. Synthesis Optimization and Characterization of Dual Drug Loaded PLGA-PEG NPs

Combo NPs, gef NPs, GSK461364A NPs, and empty NPs were synthesized and characterized for their size, PDI and zeta potential using the Zetasizer Dynamic Light Scattering instrument. Combination NP as shown in Table 1, were synthesized initially with a final concentration of polymer of 2.5 mg/mL. 

As shown in Table 1, we found that the encapsulation efficiency of the drugs in the combination NP was poor and we further optimized the polymer concentration to accommodate and entrap required concentration of each of the drugs based on the cytotoxicity test results with the free drug combinations. We synthesized the combination NPs by adding gefitinib and GSK461364A in the ratio of 4.54:1 and characterized them as shown in Table 2.

Successful encapsulation of gefitinib and GSK461364A with higher concentrations were achieved with 10 mg/mL concentration of polymer. We obtained different molar ratios of drugs encapsulated in the NPs upon characterization across various batches. Higher ratios of gefitinib: GSK461364A were obtained when multiple batches of synthesis were combined and characterized. We classified these into different groups as tabulated in Table 3.

#### Morphological Characterization

Combo NPs were characterized for their size and shape using TEM. In our previous study, we found the GSK461364A NPs to be spherical in TEM and scanning electron micrographs showed that the NPs were distinct and not clustered. As shown in Figure 2, we observed that gef NPs were spherical and the clustered when observed under SEM, with the size comparable to that measured using dynamic light scattering (DLS). Combo NPs were spherical and non-aggregated, and the particle size was similar to the sizes observed by DLS. Typically, sizes obtained by electron microscopy are smaller than that observed by DLS since DLS measures hydrodynamic size while electron microscopes give a snapshot of particles under dry conditions. 

### 3.2. In Vitro Stability Study

In vitro stability of combo NPs was evaluated by storing the samples at 4 °C and 37 °C. As shown in Figure 3, we found that the samples stored at physiologically relevant temperature had a gradual increase in size over seven days with a total increase of ~110 nm. PDI increase follows the size pattern, with a drastic increase from day 1 to day 2 with an overall increase of 0.175. Zeta potential was maintained around −20 mV and the surface charge did not become neutral until day 7, meaning that the formulation is stable at 37 °C, without drastic changes in the physicochemical properties.

Samples stored at 4 °C were much more stable as expected from our previous study, with minimal increase in size, PDI, and zeta potential thus increasing the shelf life. This proves that the formulation is stable at 4 °C and 37 °C.

### 3.3. Stability of Free vs. Nano Drug in Media Containing Fetal Bovine Serum

The stability of drugs in serum containing media was monitored over 48 h by measuring drug absorbances in free form as well as in encapsulated forms and the decrease in concentrations were plotted against time as shown in Figure 4.

As shown in Figure 4A, free gef exhibited a drastic degradation in media while nano gefitinib was much more stable at the end of 48 h with a statistical significance of *p* < 0.001. When the samples were observed after 24 h, free gefitinib precipitated while gef NPs remained stable. Figure 4B shows that free GSK461364A maintained a steady absorbance till the end of the study while nano GSK461364A exhibited increase in absorbance, which is explained by the drug release from the polymeric nano carrier. Figure 4C suggests that gefitinib and GSK461364A were released from combo NPs causing an increase in drug absorbance until 2 h and gradual decrease until 48 h. The formulation was found to be stable until 48 h. PEG increases the steric stabilization of NPs and reduces the protein adsorption present in the media [44]. This is backed by several studies that have proved PEG to drastically minimize the adsorption to proteins or even opsonization in vivo [45,46].

Overall, this experiment suggests that the nanoparticles prevent rapid drug degradation in serum causing an increase in bioavailability of the drugs and helps in preserving the structure and activity of the small molecules.

### 3.4. Measurement of In Vitro Drug Release from the Combination Nanoparticles

In vitro drug release of gefitinib and GSK461364A were evaluated by performing the dialysis bag diffusion technique under sink conditions [47]. The samples were periodically collected and since the wavelengths of the absorbances of both the drugs were close, reverse phase HPLC was used to separate the peaks based on polarity. As shown in Figure 5, we obtained three peaks at 8.309 min, 8.722 min, and 14.935 min. Based on our control (empty NPs), we established that the peak seen at 14.935 min in the chromatogram was due to the polymer.

To further evaluate and confirm each of the peaks arising from gefitinib or GSK461364A, peak eluents were fed into LC-MS system. As shown in Figure 6, peak at 8.309 min (A) had a molecular weight of 447 k.Da confirming the presence of gefitinib. The peak at 8.722 min (B) had a molecular weight of 544 k.Da confirming GSK461364A and the peak at 14.93 min (C) had various peaks as a representation of various components of co-polymers.

From this study, we confirmed that the signal from gefitinib does not interfere with the neighboring peak of GSK461364A. Once the molecular weights were confirmed, we used the area under the curve values to determine the rate of drug release with time. 

As shown in Figure 7A, 94% of gefitinib was released from combination NPs within 7 h, with 50% of release at ~75 min and 80% of drug release at 4 h. The best fit curve for the release profile follows Weibull data dissolution model with the coefficient of determination of 0.9966 and is represented by the equation
F = Fmax × {1 − Exp[−(t^β)/α]} (3)
with values of the scale parameter (α) as 1.304, the shape parameter (β) as 0.595, and Fmax as 100.737. α represents the time interval that it takes for 63.2% of the drug to be released [48]. F represents dissolution of drug as a function of time whereas Fmax is the total amount of drug released [49]. β < 1 shows that the shape of the curve is steep, as shown in Figure 7A.

Figure 7B represents the release profile of GSK461364A from combo NPs. As seen, 91% of GSK461364A was released within 7 h with a 50% of drug release at 57 min and 80% release at 4 h. The best fit curve of this release profile follows Makoid-banakar model with an R^2^ value of 0.999 and is represented by the equation
F = k_MB_ × (t − Tlag)^n × Exp[−k × (t − Tlag)](4)
where the values of empirical parameters are: k_MB_ = 56.859, n = 0.308, k = 0.018 and Tlag = 0.190. n value is less than 0.5 showing that the mechanism behind the drug release was purely diffusion [48].

Both the models are a good fit for gefitinib and GSK461364A with R^2^ value > 0.995.

### 3.5. Measurement of Cytotoxicity -Evaluation of the Treatment for Synergy in Monolayer Cultures of U87-MG

The percentages of viable cells were evaluated by treating the cells with different combination NPs with varied ratios of gefitinib:GSK461364A. To evaluate the effect of combinations on cell viability, the concentrations of the two drugs were chosen so that neither of the drugs showed a lot of cytotoxicity by themselves. As shown in Figure 8, the combination of free drugs always had a higher cytotoxicity in U87-MG compared to any of the single drugs across all concentrations. With increasing ratios of gefitinib, we observed higher cytotoxicity compared to the free drug combination. With multiple repeats and different cocktail mixtures of both the small molecules in a single nanocarrier, we concluded a crucial finding that higher ratios of gefitinib:GSK461364A are required to observe synergistic killing of U87-MG cells. The co-encapsulation strategy always worked better than combination of two single drug loaded NPs, sometimes resulting in a statistically significant killing (*p* < 0.05, *p* < 0.005) as observed in 7:1 and 9:1 ratiometric batches. Student’s *t*-test was used to determine the statistically significant values.

In addition to the Student’s *t*-test, we performed ANOVA single factor test to evaluate statistically significant values among free drug combination, nano gefitinib + nano GSK461364A, and combo NPs. We obtained F-value > F critical for combination NPs of ratios 7:1 and 9:1 suggesting statistical significance. This confirms the statistical results from the Student’s *t*-test. 

The second crucial finding from this experiment is that the drugs when in combination and encapsulated in the same carrier (combo NPs), exhibited higher cytotoxicity compared to each of the drugs in two different NPs such as gef NP and GSK461364A NP. This is due to the delivery of both the drugs from a single nanoparticle to the same site and paralyzing the cell survival, instead of non-uniform distribution of the drugs when loaded in two different drug carriers. Due to the parallel and uncoupled targeting of two drugs to two different cellular sites that trigger apoptosis, cell death is accelerated leading to a superior cytotoxic effect. We further evaluated the coefficient of drug interaction and its effect on malignant glial cells as shown in Figure 9 using the formula,
Coefficient of drug interaction (CDI) = Ratio of combination treatment to the control/((Ratio of gefitinib to control) × (Ratio of GSK461364A to control))(5)

Across all the ratios, as previously mentioned, we observed that with free drug combinations and combo NPs, we observed synergistic values (CDI < 1). As the ratio of gefitinib: GSK461364A increased, CDI values were less than 1 for combination of two single loaded NPs (gef NPs + GSK461364A NPs). Overall, we found that a strong synergism was achieved with combo NPs compared to free gef/GSK461364A and gef NPs/GSK461364A NPs. The cytotoxicity assay confirmed that with higher ratios of gefitinib, a strong synergistic activity was accomplished with combo NPs compared to the rest of the treatment arms.

### 3.6. Clonogenic Assay

Clonogenic assay was performed to estimate the clonogenic potential of the cells post combination treatment and compare this to that of the free drug and the single free drug/NP treatment. It is an assay that can assess the effects of the treatment for a longer period than the cytotoxicity assay and also helps evaluate the tumor initiating ability of the cell post treatment [50]. As shown in Figure 10A, the same number of colonies were observed in cells treated with empty NPs as no treatment, implying that the PLGA-PEG carrier did not cause any cytotoxicity to the cells.

Fewer colonies were observed in those treated with 35 µM of free gef and gef NP while no colonies in those treated with 5 µM of free GSK461364A and GSK461364A NPs. This reinforces the trend observed in cytotoxicity assay, that GSK461364A is cytotoxic at chosen concentrations with no clonogenic stimulating potential. As expected, no colonies were found in the combo NPs and their free equivalents. In the cells treated with ratiometric combo NP containing 45 µM gefitinib and 5 µM GSK461364A, no colonies were found as shown in Figure 10C. Gefitinib treatment at 45 µM in free and nano form, resulted in slightly higher cell survival rate than 35 µM treatment. 

## 4. Discussion

Numerous studies have reported different ratiometric approaches such as conjugating two drugs using a spacer [51], codelivery of multiple drugs that are loaded by conjugating to a co-polymer bound to cyclodextrins [52] etc. In our study, we have demonstrated a successful ratiometric synthesis and encapsulation of several different desired ratios of gefitinib and GSK461364A in PLGA-PEG nanoparticles and observed their synergistic effect on glioma cells. Nanoparticles have been chosen as efficient drug delivery carriers for the combination treatment using ratiometric control to treat various cancers such as bladder cancer [53], prostate cancer [54], breast cancer [55], lung carcinoma, and melanoma [56]. Research studies pertaining to GBM, have demonstrated various forms of nanoparticles as those encapsulating single chemotherapeutic agent or non-viral gene delivery vectors, conjugating the surface of the particles with a monoclonal antibody, conjugating a drug to the nanoparticles or using a photodynamic therapy (PDT) [57]. Current research also includes free drug combinations such as gefitinib and 17-N-allylamino-17-demethoxygeldanamycin (17-AAG) inhibitor [58], All-trans retinoic acid (ATRA) and interferon-γ [59], β-elemene and gefitinib [60], and nanoparticle based drug combinations such as temozolomide and bromodomain inhibitor (JQ-1) [61] for the treatment of GBM. 

The specific combination of the two drugs used in this study has never been reported earlier for treating GBM or even other forms of cancer. With our experimental results, we proved that the combination therapy is better than the monotherapy in terms of cytotoxicity as well as the clonogenic potential of glioma cells. Successful encapsulation of the drugs at various ratios were obtained and these ratiometric particles were tested for their therapeutic efficacy against GBM cells. Synthesized combination NPs were sub-200 nm in size with a zeta potential of around −20 mV, making the formulation stable and suitable for in vivo testing. The stability tests also proved that the particles were stable at body temperatures in serum until day 7, and in vitro release experiments proved that both the drugs were released from the nano carrier within 7 h. The release profiles of both the drugs from PLGA-PEG NPs are similar making it a desired drug delivery system that is taken up by the cells after which the NPs release the drugs at the same time resulting in simultaneous cytotoxic effects in glioma cells. The synthesized nano carriers proved very effective in demonstrating that they can safeguard each of the drugs from precipitating or losing its therapeutic activity when it interacts with serum. In vitro cytotoxicity studies show that higher concentrations of gefitinib in free or nano form along with GSK461364A led to strong synergistic action, while at lower concentrations of gefitinib, the cytotoxic effect of combination NP was almost the same as the free formulation as well as the cocktail mixtures of two single drug loaded PLGA-PEG NPs. The synergistic effect observed with the combination was due to the parallel action of each of the drugs that resulted in enhanced cytotoxicity. This nanoparticle mediated co-encapsulation strategy will be a great way to alter the ratio of drugs and encapsulate required doses of drugs and avoid the toxicity issues often seen in free combination or single drug treatment in sequential therapies in the clinic [25]. 

Co-encapsulated NPs have exhibited significantly higher cytotoxicity (*p* < 0.05, *p* < 0.005) in U87-MG compared to two different single drug containing NPs. This is due to the release and distribution of both drugs from the same carrier to the tumor site. If this effect was seen in monolayer cell killing, we expect to see a drastic difference when tested in vivo, due to higher entrapment of NPs in the tumor vasculature and release of the drugs. We observed a higher number of viable cells subjected to free and nano gefitinib, since the action of gefitinib is cytostatic. We noticed little to no effect of gefitinib on cell viability when assessing its cytotoxicity. However, using nano constructs will prevent the rapid metabolism of gefitinib by CYP3A4 in GBM patients [62] and increase the drug bioavailability. When in combination, due to the cytotoxic action of GSK461364A and the cytostatic action of gefitinib, we observed no viable cells in the dish. Whereas in the cells that were dosed with cocktail mixtures of single drug loaded NPs, we consistently observed a few viable cells, but no colonies. This means that a few tumor cells survived despite the exposure to both the drugs, with a possibility to form colonies after a few days. Simultaneous delivery of two drugs using polymeric nanoparticles has shown improved treatment responses especially when compared to free and individually loaded nano combinations [54,63]. The drug combinations in nanoparticles have been tested in various treatment strategies in different types of cancers such as a combination of photothermal therapy and anti-CTLA4 (cytotoxic T-lymphocyte-associated antigen 4) encapsulated in PLGA nanoparticle for immunotherapy [64], lipid/calcium/phosphate nanoparticles containing Vascular endothelial growth factor (VEGF) small interfering RNA (siRNA) and gemcitabine monophosphate to treat NSCLC [65], combination of Avastin and BPD (Verteporfin) in liposomes to treat pancreatic cancer [66].

The results from this study set a strong basis for future testing of formulations in 3D tumor models and in vivo. Future work will include testing the formulation in non-malignant cells and comparing the efficacy of nano formulations to their free equivalents. Assays to evaluate the effect of hepatic enzymes (ex: CYP3A4) on these formulations and assess their cytotoxic profiles in U87-MG can be done to prove that nanoparticles prevent rapid degradation of the drugs and also predict the in vivo pharmacokinetics [67]. In vivo work will provide a better understanding of how NPs can circumvent the toxicity issues usually seen with the free drugs and improvement in circulation half-life. This in vitro success will be effective in clinical models when the pharmacokinetics of both the drugs are carefully studied while escalating it to a clinically-relevant dose [68].

## Figures and Tables

**Figure 1 jcm-08-00367-f001:**
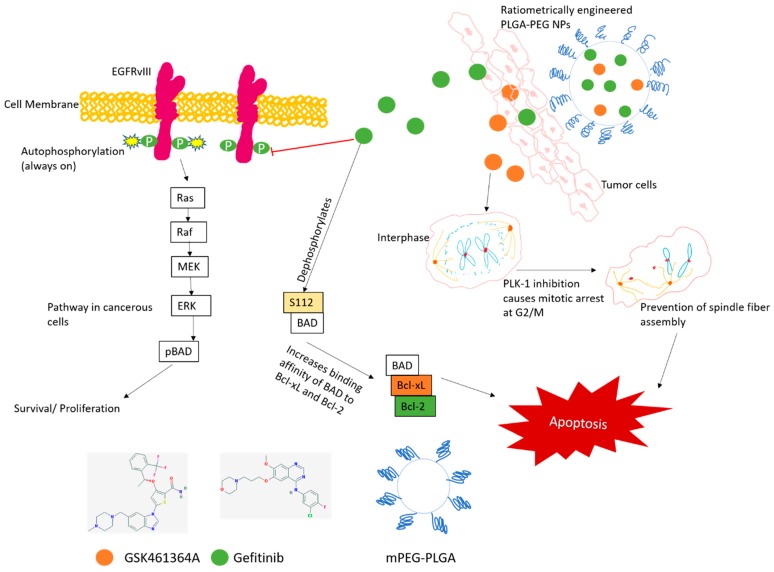
Proposed mechanism of action of dual drug loaded poly (lactic-co-glycolic) acid and polyethylene glycol (PLGA-PEG) nanoparticles and their effect on U87-MG cells. The chemical structure of gefitinib and GSK461364A were obtained from Pubchem. NPs: Polymeric nanoparticles; EGFR: epidermal growth factor receptor; BAD: Bcl-2 associated death promoter; MEK: Mitogen activated protein kinase kinase; ERK: Mitogen activated protein kinase.

**Figure 2 jcm-08-00367-f002:**
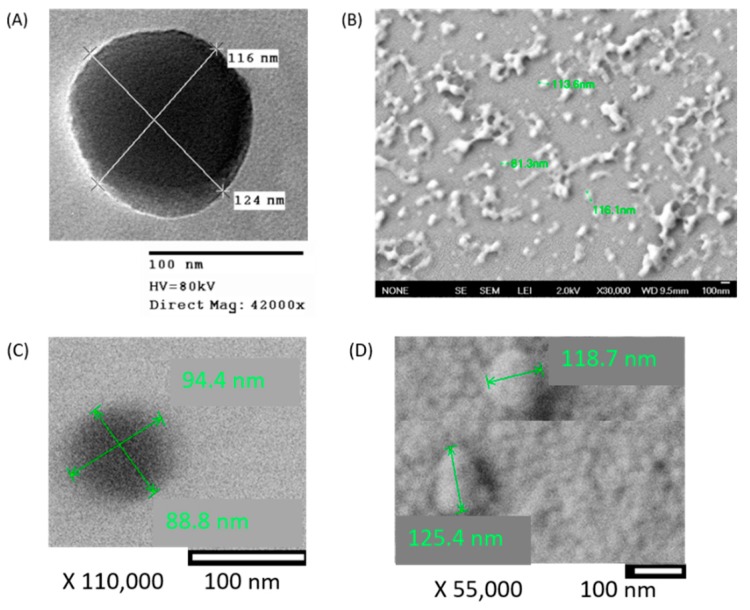
Transmission and scanning electron micrographs of drug encapsulated PLGA-PEG nanoparticles. (**A**) Transmission electron microscopy (TEM) of gefitinib in PLGA-PEG NPs. HV: High Voltage. (**B**) Scanning electron microscopy (SEM) of gefitinib in PLGA-PEG NPs (**C**) TEM of combination nanoparticles (NPs) (**D**) SEM of combination NPs. Also seen in the background are the gold particles (small raised structures) that are sputter coated on the silicon wafer.

**Figure 3 jcm-08-00367-f003:**
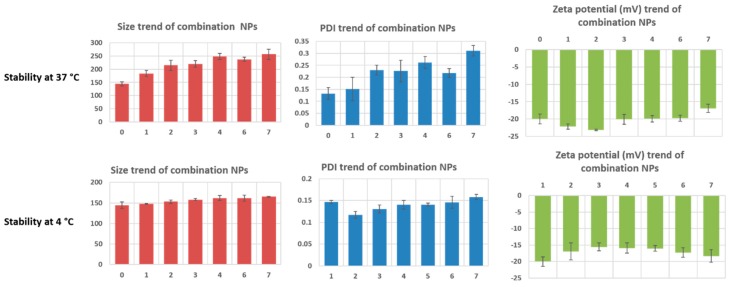
In vitro stability of combination NPs at 4 °C and at 37 °C in media containing 10% fetal bovine serum (FBS). PDI: polydispersity index.

**Figure 4 jcm-08-00367-f004:**
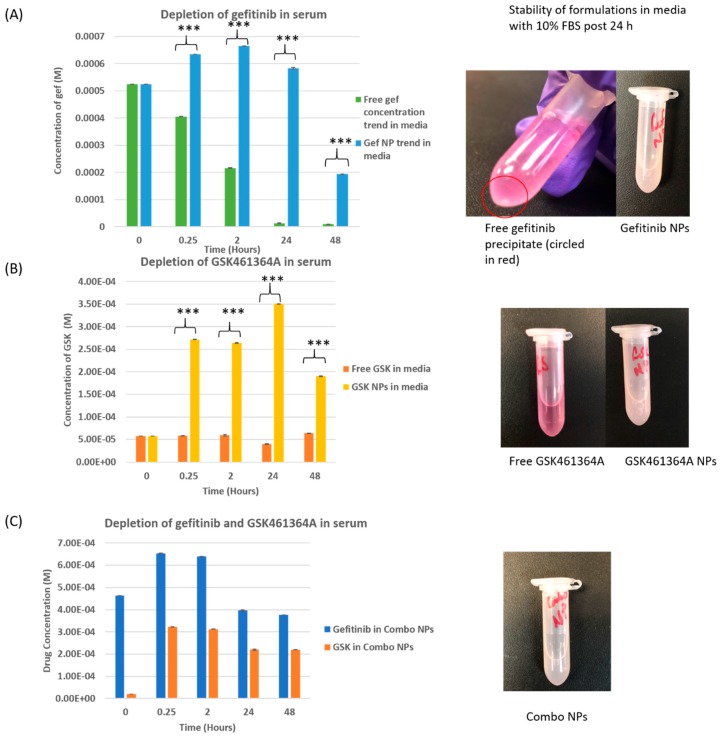
In vitro drug degradation of free gef, gef NPs, free GSK461364A, GSK461364A NPs, and combo NPs at 37 °C in media containing 10% FBS over 48 h. *** *p* < 0.001.

**Figure 5 jcm-08-00367-f005:**
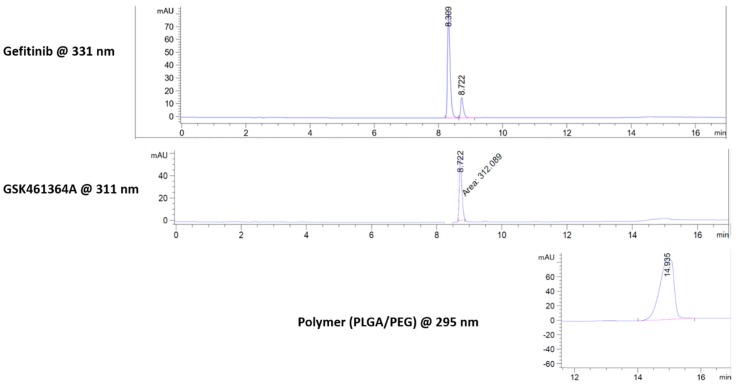
Peak separation of gefitinib, GSK461364A, and polymer (PLGA-PEG) observed in reverse phase HPLC.

**Figure 6 jcm-08-00367-f006:**
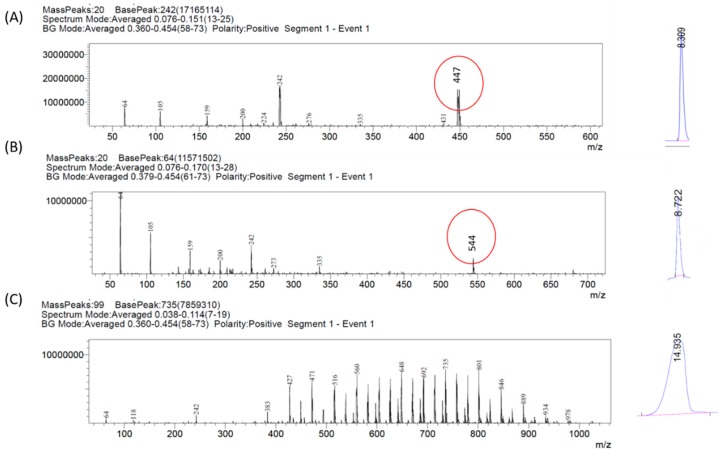
Confirmation of molecular weights of peak eluents using a mass spectrometer showing the presence of (**A**) gefitinib, (**B**) GSK461364A, and (**C**) polymer (PLGA-PEG).

**Figure 7 jcm-08-00367-f007:**
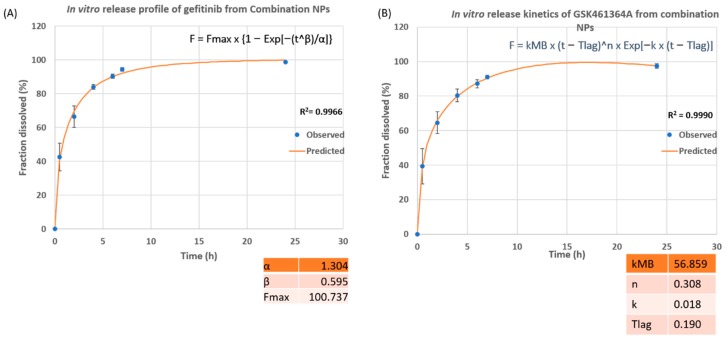
In vitro release profile of (**A**) gefitinib and (**B**) GSK461364A from dual drug loaded PLGA-PEG nanoparticles (combo NPs) at 37 °C, 1X PBS, pH 7.4.

**Figure 8 jcm-08-00367-f008:**
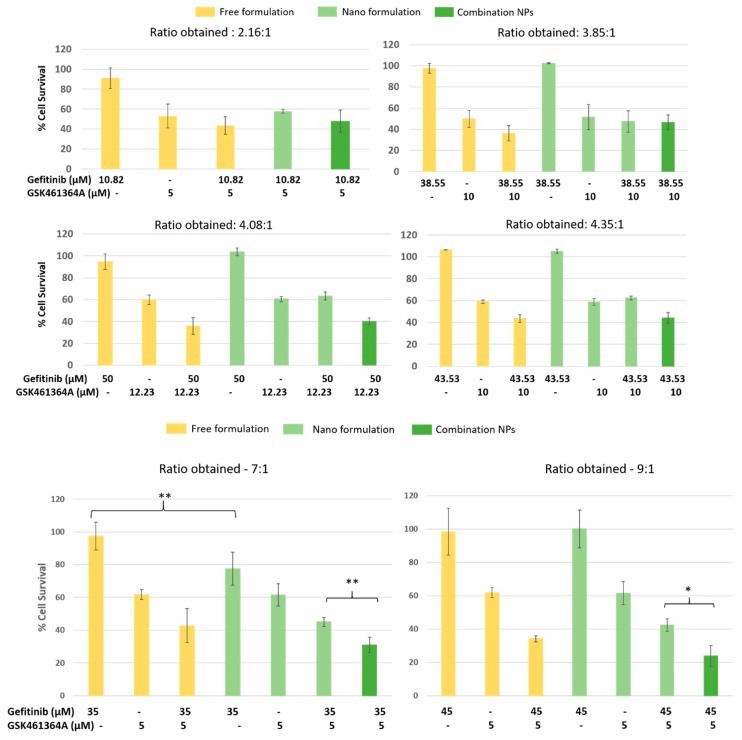
In vitro cytotoxicity assay showing difference in cell viabilities when treated with combination NPs arranged in the increasing molar ratios of gefitinib: GSK461364A. * *p* < 0.05, ** *p* < 0.005.

**Figure 9 jcm-08-00367-f009:**
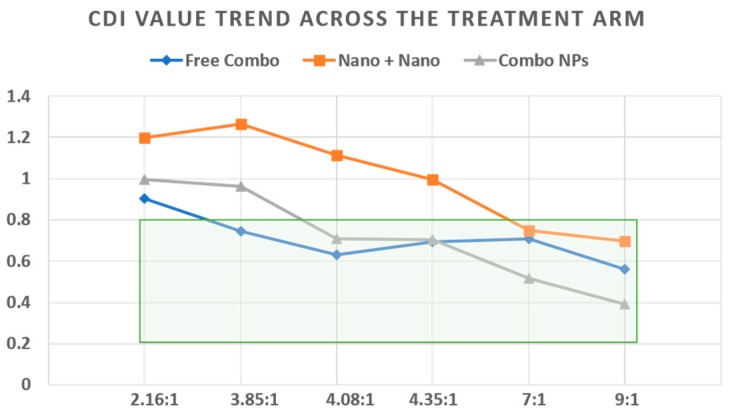
Co-efficient of drug interactions with various molar ratios of gefitinib and GSK461364A in combo NPs and their effects in U87-MG cell line. Coefficient of drug interaction (CDI) values > 1 antagonistic, =1 additive, <1 synergistic (green box).

**Figure 10 jcm-08-00367-f010:**
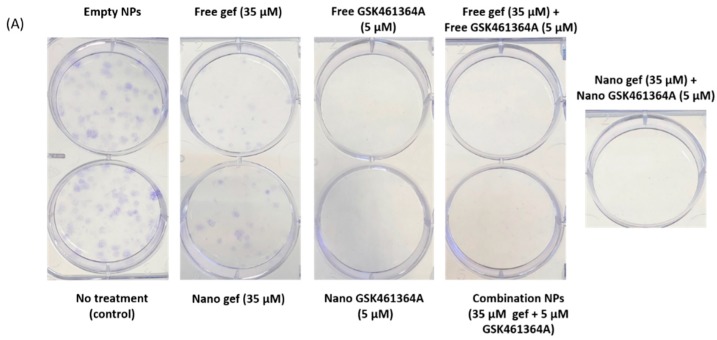
Colonies formed by U87-MG cells post treatment with combination NPs of different concentrations, single drug loaded NPs, respective free drug equivalents and control (**A**,**C**); Quantified cell colony formation data with the error bars indicating standard deviation of three repeats (**B**,**D**). NT: No Treatment.

**Table 1 jcm-08-00367-t001:** Characterization of empty, single drug, and dual drug loaded PLGA-PEG NPs.

Nanoparticle Type	Mean Hydrodynamic Size (nm)	PDI	Zeta Potential (mV)	Molar Ratio of Gefitinib: GSK461364A	% Encapsulation Efficiency	
PLGA-PEG (empty NPs)	127.0 ± 0.3464	0.097 ± 0.005	−35.5 ± 5.90	N/A	N/A	
Gef NPs	116.8 ± 1.62	0.120 ± 0.6576	−23.4 ± 0.707	N/A	68.34 ± 6.438	
GSK461364A NPs	120.5 ± 2.483	0.115 ± 0.004	−25.6 ± 0.451	N/A	56.29 ± 4.319 [37]	
Combo NPs	101.29 ± 2.412	0.102 ± 0.034	−24.65 ± 1.943	1:1	26 ± 8.98 gefitinib	35 ± 5.34 GSK461364A

PLGA-PEG: poly (lactic-co-glycolic) acid and polyethylene glycol; NPs: Polymeric nanoparticles; PDI: polydispersity index.N/A: Not applicable.

**Table 2 jcm-08-00367-t002:** Polymer concentration optimization for successful synthesis and encapsulation of gefitinib and GSK461364A in PLGA-PEG NPs.

Final Polymer Concentration (mg/mL)	Mean Hydrodynamic Size (nm)	PDI	Average Zeta Potential (mV)	% Encapsulation Efficiency	
				Gefitinib	GSK461364A
2.5	Synthesis Unsuccessful				
5	408.25 ± 48.98	0.239	−24.5 ± 0.458	6.65 ± 4.45	None
10	101.29 ± 2.412	0.133	−29.6 ± 0.777	70.11± 6.33	65.74 ± 10.85

**Table 3 jcm-08-00367-t003:** Characterization features of engineered NPs using ratiometric control.

**Molar Ratio of Gefitinib:GSK461364A**	(2.16–3):1	(3.6–4.35):1	7:1	9:1
**Encapsulated gefitinib (µM)**	1146.88	1959.26	1294.79	590.51
**Encapsulated GSK461364A (µM)**	386.28	508.18	184.27	66.38
**Size (nm)**	160 ± 21.385	141.04 ± 25.156	142.8 ± 14.153	129.75 ± 22.69
**PDI**	0.15 ± 0.0487	0.146 ± 0.052	0.168 ± 0.044	0.154 ± 0.045
**Zeta potential (mV)**	−20.44 ± 3.069	−20.867 ± 4.055	−19.78 ± 2.252	−20.25 ± 7.848

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
