# Peer review of "Synergistic Action of Gefitinib and GSK41364A Simultaneously Loaded in Ratiometrically-Engineered Polymeric Nanoparticles for Glioblastoma Multiforme"

_jcm, 2019, doi:10.3390/jcm8030367_

Round 1
Reviewer 1 Report
This is an interesting paper. The Introduction should be rewritten to clearly state the background on the disease, the background on the nanotechnology and the drugs, and why this work is needed. It should be checked whether ANOVA instead of t-test should be used for the statistics.
Author Response
We thank the reviewer for his positive feedback. We have rewritten the background as described by the reviewer. We also have now included ANOVA-based statistical comparisons for the cytotoxicity results.

Reviewer 2 Report
This is valuble and well-done work.
This work presents new and important data as well as a concept that, while not new is generally underappreciated. Combining synergic molecules with different MOAs within a nanoparticle is a major advance as a general concept and the authors’ specific combination data should be more widely known.
Title is needlessly obscure. Both drugs must be mentioned in title specifically.
A paragraph or two in the Introduction must be devoted to GSK461364A, giving basic pharmacology parameters, MOA or putative MOA, and an overview of polo-like kinases’ role in normal cell cycling and in GB. The entire Introduction requires a rewrite.
Fig.1 must be corrected. Having an inhibition “T” arrow from an orange circle to the cell cycle circle is unacceptable. The details must be shown. Likewise unacceptable is the “dephosphorylation” arrow. By what means dephosphorylation ? Does gefitinib really dephosphorylate BAD or rather does gefitinib inhibit BAD phosphorylation ? The difference is not trivial. In other word much detail must be added to Fig. 1 for it to become meaningful.
Line 41 on pg.1 must be corrected. “hostile effects” is unacceptable English use in this context
If the authors could organize their Introduction better, intelligibility would be served. Ideally a paragraph devoted to PLK, another paragraph to EGFR and gefitinib, and another to the very important concept of simultaneously interfering with multiple non-parallel targets in the dysregulated cancer signaling pathways as more effective than sequential treatments. This latter requires its own paragraph and discussion with delineated example[s].
It would be preferable to show the graphic output of the Malvern Zetasizer 121 Nano ZS90.
Given the importance of nanoparticles’ attributes to the authors’ work, a separate paragraph explaining why nanoparticles might get better intracellular uptake than would ordinary soluble molecules would enhance this paper’s value.
I would prefer to see “GSK461364A” used throughout the paper, not “GSK”. GSK is the well-known abbreviation of a drug company.
The authors’ lab work seems well done, appropriate to support their conclusions.
Reviewer 3 Report
Dear authors,
the search for new and better active substances or combinations of active substances for the therapy of glioblastoma is still of great relevance. The combination of active compounds you have chosen, packaged in nanoparticles, sounds very promising. In the introduction, some aspects could be examined in more detail. These include the current studies on combination therapy with TMZ and CCNU and, of course, TTF therapy, which leads to a significantly improved survival advantage for patients. This study (EF-14) should also be cited in line 36/37, no data from 2012. A literature search on the BBB mobility of the selected particles would also be important. Are there any drug-drug interactions between gefitinib and GSK461364? In line 45, the sentence should be reformulated to cell cycle arrest.
In the methods section, the exact device designations and manufacturers should be listed for the devices used. The formulas should be numbered. In section 2.5, line 175, the expression " some cells" should be specified. In addition, the source for the determination of the EC50 should be indicated by GSK.
I would also like to ask the authors to check the spelling in the manuscript. Latin phrases are written in italics.
In addition, I would like to suggest for future analysis by the authors that the investigations should not only be performed on commercial cell lines, but also on primary adherent and spheroid cultures derived from surgical material.
Author Response
Please see attached file for our point-by-point response to the reviewers's comment.
